# NSAID-Induced Enteropathy Affects Regulation of Hepatic Glucose Production by Decreasing GLP-1 Secretion

**DOI:** 10.3390/nu14010120

**Published:** 2021-12-28

**Authors:** Hussein Herz, Yang Song, Yuanchao Ye, Liping Tian, Benjamin Linden, Marwa Abu El Haija, Yi Chu, Justin L. Grobe, Randall W. Lengeling, Mohamad Mokadem

**Affiliations:** 1Department of Internal Medicine, Roy J. and Lucille A. Carver College of Medicine, University of Iowa, Iowa City, IA 52242, USA; hussein-herz@uiowa.edu (H.H.); songyanglhyb1998@163.com (Y.S.); yuanchaoye1986@gmail.com (Y.Y.); lipingtian2019@gmail.com (L.T.); benjamin-linden@uiowa.edu (B.L.); yi-chu@uiowa.edu (Y.C.); 2Stead Family Department of Pediatrics, University of Iowa Carver College of Medicine, Iowa City, IA 52242, USA; Marwa.Haija@stanford.edu; 3Department of Pediatrics, Division of Gastroenterology, Hepatology, and Nutrition, Stanford University School of Medicine & Lucile Packard Children’s Hospital, Palo Alto, CA 94304, USA; 4Department of Physiology, Medical College of Wisconsin, Milwaukee, WI 53226, USA; jgrobe@mcw.edu; 5Comprehensive Rodent Metabolic Phenotyping Core, Medical College of Wisconsin, Milwaiukee, WI 53226, USA; 6Cardiovascular Center, Medical College of Wisconsin, Milwaukee, WI 53226, USA; 7Neuroscience Research Center, Medical College of Wisconsin, Milwaukee, WI 53226, USA; 8Gastroenterology Section, Grand River Medical Group, Dubuque, IA 52001, USA; rlcl1165@aol.com; 9Fraternal Orders of Eagles Diabetes Research Center, University of Iowa, Iowa City, IA 52242, USA; 10Obesity Research & Education Initiative, University of Iowa, Iowa City, IA 52242, USA; 11Veterans Affairs Health Care System, Iowa City, IA 52242, USA

**Keywords:** non-steroidal anti-inflammatory drugs (NSAID), glucagon-like peptiode-1 (GLP-1), NSAID-induced enteropathy, hepatic glucose production, hepatic insulin sensitivity, glucose tolerance

## Abstract

Background/Aim: Given their widespread use and their notorious effects on the lining of gut cells, including the enteroendocrine cells, we explored if chronic exposure to non-steroidal anti-inflammatory drugs (NSAIDs) affects metabolic balance in a mouse model of NSAID-induced enteropathy. Method: We administered variable NSAIDs to C57Blk/6J mice through intragastric gavage and measured their energy balance, glucose hemostasis, and GLP-1 levels. We treated them with Exendin-9 and Exendin-4 and ran a euglycemic-hyperinsulinemic clamp. Results: Chronic administration of multiple NSAIDs to C57Blk/6J mice induces ileal ulcerations and weight loss in animals consuming a high-fat diet. Despite losing weight, NSAID-treated mice exhibit no improvement in their glucose tolerance. Furthermore, glucose-stimulated (glucagon-like peptide -1) GLP-1 is significantly attenuated in the NSAID-treated groups. In addition, Exendin-9—a GLP-1 receptor antagonist—worsens glucose tolerance in the control group but not in the NSAID-treated group. Finally, the hyper-insulinemic euglycemic clamp study shows that endogenous glucose production, total glucose disposal, and their associated insulin levels were similar among an ibuprofen-treated group and its control. Exendin-4, a GLP-1 receptor agonist, reduces insulin levels in the ibuprofen group compared to their controls for the same glucose exchange rates. Conclusions: Chronic NSAID use can induce small intestinal ulcerations, which can affect intestinal GLP-1 production, hepatic insulin sensitivity, and consequently, hepatic glucose production.

## 1. Introduction

Nonsteroidal anti-inflammatory drug (NSAID) damage to the small intestine was first described in a rat model using indomethacin in 1969, while a case report of small bowel ulcers in two patients taking the same medication was reported three years earlier [1,2]. However, for the next quarter century, these reports largely escaped clinical attention until studies and reviews effectively validated enteropathy with nearly all NSAIDs, including aspirin [3,4,5,6,7] and selective cyclo-oxygenase-2 (COX-2) inhibitors, which were originally designed to prevent upper GI injury [7,8]. Though NSAID damage occurs throughout the small bowel, distal small intestinal damage may be more severe for several reasons. These include colonic reflux of bacteria into the terminal ileum (as bacteria have been suggested to play a role in this drug-induced insult), as well as the ileal brake and enterohepatic circulation of the drug [9]. The first NSAID, essentially acetylsalicylic acid (ASA), has been available since the beginning of the 20th century and, for decades, has been available over the counter (OTC). Non-aspirin NSAIDs were available by prescription since the middle of the past century, but it was not until 1984 that they became available OTC in the United States. Currently, there are more than 70 million prescriptions for NSAIDs that are written each year, and an estimated 30 billion doses of NSAIDs are consumed annually in the United States alone [10].

The L-cells are entero-endocrine secretory cells lining the mucosa of the luminal gut and are more abundant in the distal gastrointestinal tract (i.e., the ileum and right side of the colon). These cells have microvilli and are in direct contact with the luminal gut, where they play a vital role in nutritional sensing and regulating gut motility and integrity, as well as the host’s overall energy and glucose metabolism by secreting hormones, such as glucagon-like peptide-1 (GLP-1), GLP-2, oxyntomodylin, and peptide-YY(PYY) [11]. These cells are superficially located between mucosal epithelial and, therefore, they are at high risk for damage from NSAID-induced injury. GLP-1 has been shown to be a key regulator of glucose homeostasis by increasing post-prandial insulin secretion through its incretin effect and by autonomically regulating endogenous hepatic glucose production [12]. It also promotes satiety and decreases appetite, increases lipogenesis and slows gastrointestinal motility, among its many other effects [13]. In fact, several GLP-1 receptor agonists are currently being used as new pharmacotherapy lines for weight loss and diabetes control [14,15,16,17]. Given the fact that NSAID-induced enteropathy has a propensity for the distal ileum, the superficial L-cells are theoretically and likely at risk for damage and dysfunctional secretory activity and, as a consequence, may lead to metabolic derangements, such as diabetes, obesity, or fatty liver disease.

To test this hypothesis, we developed and confirmed a murine model of NSAID-induced enteropathy by exposing mice to multiple NSAID formulations (at physiologic doses) for several weeks. We then examined their body weights, calorie consumption, energy expenditure, glucose and insulin tolerance, and GLP-1 stimulated secretion compared to control mice while on two different diets. Finally, we tested if GLP-1 receptor agonist and/or antagonist manipulation would affect some of the observed metabolic variables.

## 2. Materials and Methods

### 2.1. Animal Care

All animal care and procedures have been approved by the University of Iowa Animal Care and Use Committee.

### 2.2. Study Design

A group of one hundred 10-week-old C57black/6J male mice were bought from (Jackson laboratory, Bar Harbor, ME, USA; Sacramento, CA, USA). They were allowed to acclimate for two weeks then one group of 50 mice was placed on a normal chow diet (ND) (59% carbohydrates, 23% protein, 18% fat), and another group of 50 mice was placed on a high-fat diet (HFD) (60% calories from fat D12492, Research Diets). Each diet group was divided into several NSAID-receiving groups and one placebo-receiving group. The treatment of ibuprofen, naproxen, indomethacin, or placebo was delivered to the distal stomach by intra-gastric gavage with a special blunt needle device (see experiment below). An initial group was sacrificed after 2 weeks of NSAID administration to examine and confirm the NSAID-induced enteropathy model by immunohistochemistry on multiple sections of the small intestine (duodenum, jejunum, and ileum). Then, two groups of animals were placed on either a normal chow or a high-fat diet (as mentioned above), and each of them received multiple NSAID formulations vs. placebos for 6 weeks. During this time period, total body weight, caloric intake, intestinal caloric absorption, total energy expenditure, glucose tolerance tests, and insulin tolerance tests were performed or calculated as described below. Finally, serum GLP-1 levels were measured in the fasting state and after oral glucose gavage (1 g/Kg), after which all animals were sacrificed, and tissues were collected. The duodenum was identified anatomically starting from the pylorus, and the 5 cm proximal was resected and cleaned with normal saline; the jejunum was identified anatomically starting from the ligament of Treitz, and the 5 cm proximal was resected and cleaned with normal saline; the ileum was identified anatomically by connection to the cecum, and the 5 cm distal was resected and cleaned with normal saline. Some sections of the small intestines were embedded in OCT for IF staining and some in paraffin for H & E staining. The rest of the tissues were snap frozen and stored in −80°.

### 2.3. NSAID Gavage Experiment

Mice aged 12 weeks old were divided into two treatment groups: a control group gavaged with normal saline and multiple NSAID groups gavaged with ibuprofen (200/mg/Kg), naproxen (100 mg/Kg), or indomethacin 20 mg/Kg. We gavaged mice every other day for a 2-week period. The procedure of gavaging consisted of grasping the mouse and holding it firmly to immobilize it with the head and body held vertically, followed by gentle sliding of a 4 cm stainless steel needle with a 4 mm blunted tip into the distal stomach of the mouse and gavaging the content of the syringe. Two separate groups of placebo vs. ibuprofen-treated mice were generated also for the GLP-1 antagonist study and the hyperinsulinemic-euglycemic clamp study, as mentioned below.

### 2.4. Energy Balance Calculations

Body weight was recorded weekly in all mice, who were housed singly in order to appropriately measure all elements of energy balance. Energy intake was obtained in week 2 and week 5 of the study by measuring the amount of food consumed per day (in grams) over 5 days—for each mouse—and calculating the average. The amount of energy consumed in KJ is, therefore, 13.4 KJ for each gram of the normal chow diet and 21.76 KJ for each gram of the high-fat diet. Energy expenditure (EE), defined as energy available for basal metabolic rate and physical activity, was calculated using a validated formula against calorimetry, where EE = Metabolizable Energy Intake − Changes Energy Stores [18,19,20,21]. The EE was corrected for the body weight of each mouse using ANCOVA. Metabolizable Energy intake is defined as the amount of energy absorbed and ready for use after accounting for the digestive process. This is 12.57 kJ multiplied by each gram of the normal diet consumed and 15.34 KJ multiplied by each gram of the high-fat diet consumed. Changes in energy stores account for the energy value of each g of fat and lean mass + energy cost of adding lean and fat mass. The energy cost for gaining 1 g of fat and lean mass is 55.3 KJ and 9.2 k, respectively, and the energy contents of fat and lean mass were estimated as 37.7 kJ and 4.2 kJ for each g, respectively [21,22]. Intestinal calorie absorption was calculated by subtracting calories collected from food intake over three days (during week 4) from daily calories remaining in the stool. Fecal calorie excretion was measured using bomb calorimetry (Parr).

### 2.5. Glucose Homeostasis Studies

Glucose Tolerance Test (GTT)/Insulin Tolerance Test (ITT) tests were performed. GTT was performed following an overnight fast for 16 h (4–5 weeks after initiation of NSAID treatment), mice received 1 g/kg D- Glucose (Cas No: 50-99-7, RPI, Mt. Prospect, IL, USA) by oral gavage. Blood glucose was measured from tail vein blood using a hand-held glucometer (Contour) immediately before and 15, 30, 60, 90, and 120 min after glucose administration. ITT was performed following a 4-h fast (3–4 weeks after initiation of NSAID treatment), mice received 0.75 U/kg Insulin (NDC 0002-8215-01, HI-210, Humulin R, Eli Lily) by intraperitoneal injection. Blood glucose was measured from tail vein blood using a hand-held glucometer (Contour) immediately before and 15, 30, 60, 90, and 120 min after glucose administration.

### 2.6. Insulin Measurement by ELISA

The sandwich-type immunoassay ELISA is used to measure plasma insulin levels (Crystal Chem, IL, USA, Cat# 90080). The microplate consists of 96-well coated with a monoclonal antibody specific for insulin. In addition to the samples, a detection antibody is added to the microplate wells. The first incubation is completed for 2 h at 4 °C on a microplate shaker at 700–900 rpm. Then, the wells were washed using a wash buffer and blotted dry. A conjugation solution of 100 µL was added, followed by 30 min incubation and another wash. After the addition of the 100 µL TMD substrate, the microplate is incubated a second time for 40 min on a microplate shaker at 700–900 rpm. Finally, 100 µL of stop solution is added, followed by the measurement of the optical density (OD) by a spectrophotometer at 450 nm. The amount of insulin in each sample is directly proportional to the intensity of the color generated.

### 2.7. GLP-1 Measurement Experiment

Peripheral plasma active GLP-1 was measured after an overnight fast and five minutes after oral gavage of 1 g/kg D-Glucose. Glucose was administered via a feeding tube to avoid stimulation of the taste buds of the lingual epithelium. Blood was collected into EDTA-coated Microvettes (SARSTEDT Inc., Newton, NC, USA) preloaded with an enzyme inhibitor cocktail containing 0.5 MEDTA (Fisher Scientific, Waltham, MA, USA), 283 μM aprotinin (Sigma-Aldrich, Burlington, MA, USA), 10,000 U/mL heparin (Sigma-Aldrich, Burlington, MA, USA), and 1.265 mM Diprotin A (Bachem, CA, USA). GLP-1 was measured using the Active GLP-1 (ver.2) Assay Kit from Meso Scale Discovery (Rockville, MD, USA), as previously described [23].

### 2.8. H & E Staining and Imaging

Tissues were fixed in 4% paraformaldehyde (PFA) and embedded in paraffin. The sections were deparaffinized in xylene solution and then in 100% ethanol solution, followed by rehydration with deionized water. The tissues were stained with hematoxylin solution for 6 min. at a temperature of 60–70 °C and were then rinsed in tap water. Next, sections were dipped 10× fast in acid ethanol and rinsed with tap water. Then staining was performed with eosin ethanol solution for 5 min. Finally, we used coverslip slides using Permount (xylene-based).

### 2.9. Immunofluorescence Antibody Staining

Ileum dissected from a mouse was embedded in OCT at −80 °C. Cuts of 10 μm cryosections were obtained on a microscope slide and put in room temperature for 30 min, followed by fixation in 4% PFA for 15 min. Blocking of tissue was conducted in blocking buffer containing 5% normal goat serum and 0.1% Triton x-100 in 1xPBS and incubated for 1 h at room temperature. As for the primary antibodies, rabbit anti-GLP-1 (ab22625, Abcam, Waltham, MA, USA) 1:100 dilution in blocking buffer was applied for 24 h at 4 °C. IgG Goat anti-Rabbit Alexa Fluor Plus 555 (2 mg/mL) 1:250 dilution was applied for 2 h as the secondary antibody at room temperature. The cells were counterstained with a nuclear stain VECTASHIELD DAPI mounting medium for 1 h at room temperature.

### 2.10. Confocal Imaging and Image Processing

Images were captured using a Leica SP8 scanning laser confocal microscope (inverted) equipped with 405, 488, 552, and 638 nm laser lines. Fast scanning with a resonant scanner (up to 40 frames/sec at 1024 × 1024 scan format). Adaptive multicolor super-resolution imaging with fast (near real-time) deconvolution software integrated with the hardware to optimize imaging parameters for the highest possible resolution. Objective lens 20× water immersion objective (670 um working distance) with motorized correction (MotCorr) collar to maximize brightness and resolution. All images were processed and quantified using ImageJ software to split and merge channels and distinct channels with colors.

### 2.11. GLP-1 Antagonist (Exendin-9) Experiment

One group of 12-week-old mice was placed on HFD and gavaged with ibuprofen (as described above) for two weeks. A single dose of 5 µg of exendin (9–39) (133514-43-9 from Sigma-Aldrich, Inc.) in 8% gelatin or an equivalent volume of PBS was administered by IP injection 20 min before initiation of glucose gavage, as mentioned in the Oral glucose tolerance test (OGTT) [24].

### 2.12. Hyper-Insulinemic Euglycemic Clamp Study

A large group of 12-week-old mice was also placed on HFD and gavaged with ibuprofen vs. placebo (PBS) for 2 weeks. The mice were then transferred to the Metabolic Phenotyping Core facility, where a vascular catheter was inserted into the jugular vein under anesthesia as per their protocol. Each group (NSAID vs. Placebo) was then divided into two groups. One receiving Exendin-4 (24 nmol/Kg) and one receiving (normal saline of equal volume) IP daily for 5 days before the clamp study [12]. A high euglycemic-hyperinsulinemic clamp was performed in the post-absorptive state (i.e., ~5 h fasting) at a rate of 4 mU/Kg/min in conscious mice or during terminal anesthesia. At time 0, insulin was infused intravenously. The glucose solution was concomitantly infused at an adjustable rate to maintain euglycemia at 150 mg/dL. Insulin infusion was maintained for up to 4 h, over which timed samples were collected. In some untraced clamps, 45 min prior to the conclusion of the clamp, filter-sterilized fluorescein solution (200 mg/mL) in saline was administered as a single bolus (50 uL; delivered intravenously via the bolus line in 1 min) in order to test the catheters for leaks.

### 2.13. Statistical Analysis

All data points are shown as a mean ± Standard Error of Means (SEM). *p* values less than 0.05 were considered statistically significant. Student’s *t* tests were used to compare two sets of groups, while 1- or 2-way ANOVA that was followed by Tukey–Kramer post hoc analysis was appropriately used to compare three or more sets of groups. Statistical analyses were performed using GraphPad prism 9.0.Ink (GraphPad software). ANCOVA was utilized to correct energy expenditure data for body mass using SPSS, as pointed in the figures.

## 3. Results

### 3.1. A Murine Model of NSAID-Induced Enteropathy

We gavaged 12-week-old C57Blc/6J male mice with multiple NSAIDs (indomethacin, ibuprofen, and naproxen) in their distal stomach on alternative days for two weeks and examined their small intestine for histologic injury. Hematoxylin & Eosin (H&E) stains of different sections of the small intestine (duodenum, jejunum, and ileum)-showed patchy mucosal sloughing/ulceration, crypt damage with lymphocytic infiltration in all the NSAID-treated groups but worse in the ibuprofen and naproxen groups than in the indomethacin one. This NSAID-induced enteropathy also seems more prominent in the distal small intestine (i.e., ileum) (Figure 1).

### 3.2. Prolonged NSAIDs Intake Protects against Weight Gain on HFD

Intragastric administration of variable NSAIDs over 2 weeks induced a small but significant decrease in total body weight in ND-fed mice when compared to placebo recipients or controls (Con) (Change in body weight over two weeks, Con: 1.7 ± 0.2 g, Ibp: 0.2 ± 0.6 g, Nap: 0.3 ± 0.3 g, and Indo: 0.4 ± 0.4 g). This effect seems to be transient as no significant change in total body weight was observed at 6 weeks (Figure 2A,B). Consequently, the average daily energy intake and daily energy expenditure were reduced in all the NSAID-treated groups compared to the controls after 2 weeks of NSAID exposure, but these energy balance elements were unchanged after 6 weeks of treatment in normal chow-fed mice (Figure 2C–F). On the other hand, mice who were placed on HFD showed significantly lower body weights at 2 and 6 weeks of NSAID exposure when compared to their controls (Figure 3A,B). Similar to what was observed in ND-fed mice, energy intake was reduced at 2 weeks but was unchanged at 6 weeks of intervention (Figure 3C,D). Interestingly, the early reduction in daily energy expenditure at week 2, which is likely an adaptation to the decrease in energy consumption, was also observed in the NSAID-treated HFD groups (Figure 3E). However, the NSAID-treated group displayed a relative increase in energy expenditure after 6 weeks while on HFD (Figure 3F). Finally, there was no difference in fecal caloric loss (measured using bomb calorimetry) between the NSAID and placebo-treated groups on either diet (Appendix A).

### 3.3. Mice Receiving NSAIDs Exhibit Improved Insulin Sensitivity but Not Glucose Tolerance

In order to assess the effect of chronic NSAID supplementation on glucose homeostasis, we performed insulin and glucose tolerance tests in mice fed with ND or HFD. The insulin tolerance test (ITT) showed significant improvement in all NSAID-treated animals compared to their controls for both diets (Figure 4A,B), as shown by the changes in the Area Under the Curves (AUCs). Interestingly, the glucose tolerance test (GTT) demonstrated similar blood glucose levels variations for all three NSAID groups compared to their controls on both diets (Figure 4C,D). Furthermore, the fasting plasma insulin levels of mice on both diets showed tendency towards reduction, but it was only significant for the indomethacin-treated group (plasma insulin levels on ND; Con= 242.8 ± 38.47, Ibp = 148.3 ± 4.52, Nap = 185.3 ± 21.12, Indo = 119.7 ± 13.48 pg/mL and plasma insulin levels on HFD; Con = 443.3 ± 122.6, Ibp = 239.9 ± 14.3, Nap = 218.7 ± 38.84, Indo = 155.6 ± 11.55 pg/mL) (Figure 4E,F).

### 3.4. NSAIDs Attenuates Incretin GLP-1 Secretion

To test the effect of NSAIDs- induced intestinal injury on GLP-1 (as a major incretin hormone), we examined fasting and glucose-stimulated levels of GLP-1 in all NSAID-treated groups on both diets. We found that the increase in GLP-1 levels post-glucose gavage in the control group was absent in all NSAID-treated groups in mice placed on ND (Figure 5A) and HFD (Figure 5B).

### 3.5. Exendin-9 Does Not Affect Oral Glucose Tolerance of Ibuprofen-Treated Mice While Exendin-4 Improves Their Insulin-Mediated Glucose Regulation

To further investigate the association between NSAID and glucose metabolism through GLP-1 signaling, we examined the effect of the GLP-1 receptor antagonist (Exendin-9) on GTT in ibuprofen vs. placebo-treated mice. Interestingly, Exendin-9 worsened the GTT for the placebo group but not the ibuprofen group (as shown by the AUCs) (Figure 6). Thereafter, we examined if the administration of a GLP-1 agonist corrects this NSAID-induced disruption in glucose metabolism. We performed a hyperinsulinemic-euglycemic clamp experiment in a group of mice placed on HFD and treated with ibuprofen vs. placebo for two weeks, followed by the administration of Exendin-4 (24 nmol/Kg IP daily for 5 days) or normal saline (NS) injection. The variation of insulin concentration as a function of glucose flux for Placebo + NS and ibuprofen + NS showed similar glucose appearance rate (Ra) and glucose disposal rate (Rd) curve trends (Figure 7A,B). The administration of Exendin-4 resulted in lower insulin levels for the same glucose appearance flux (Ra) only in the ibuprofen-treated animals (Figure 7A lower panel). Similarly, insulin levels were lower post-Exendin-4 for the same glucose disposal flux (Rd) in the ibuprofen-treated group compared to its control (Figure 7B lower panel).

## 4. Discussion

Animal models, almost exclusively murine, have been pivotal in the discovery and understanding of the pathogenesis of what is now called NSAID-induced enteropathy in humans, which has been corroborated by subsequent human research. As alluded to above, this first began in 1968 when rat models were designed to study the cause of peptic ulcer disease using indomethacin only to discover that small bowel ulcerations and not gastric damage was the predominant pathology [2]. Further studies in animals led to a better understanding of the pathogenesis of this drug-induced small bowel injury [25,26,27], and subsequent reports documented its high prevalence in humans later [4,5,6,9]. The pathology has been most damaging in the distal small bowel, specifically the distal jejunum and ileum, and care needs to be taken as to not diagnose these findings as incurable Crohn’s Disease [7].

Over 20 hormones are produced by the mucosal epithelial and entero-endocrine cells lining the gut and are critical in maintaining energy balance and glucose homeostasis [28]. However, amongst all of the gut-secreted hormones, GLP-1 has gained the most attention as being the most powerful incretin in addition to independently regulating peripheral and hepatic insulin sensitivity, as well as appetite and energy expenditure [12,13]. Advancement in our understanding of GLP-1 signaling and its role in glucose metabolism has led to the development of several pharmacologically active GLP-1 receptor agonists, such as liraglutide, exenatide, and semaglutide, for the treatment of type 2 diabetes [14,15,29]. Interestingly, this hormone was shown to be significantly altered in our suggested experiment, specifically post-glucose gavage, which translates into the post-prandial state or “fed-state” where GLP-1 is typically secreted. We observed that chronic administration of NSAID causes a relative state of GLP-1 depletion, likely due to the mucosal ulceration that can also affect the GLP-1 secreting L-cells which are typically abundant in the distal small intestine. We did not find any evidence that NSAIDs affect L-cells’ capability to synthesize the GLP-1 peptide as immunohistochemical staining of GLP-1 within the ileum showed no difference between the NSAID group and their controls in the fasting/non-stimulated state (Appendix A).

Remarkably, this NSAID-induced GLP-1 deficiency did not cause a major body weight shift in lean mice fed a normal chow diet but caused weight loss in mice placed on an obesogenic high-fat diet. This is consistent with published data showing that genetic mouse models of GLP-1 receptor deficiency have no weight phenotype compared to wild-type counterparts on a regular diet but shows resistance to weight gain when placed on a high-fat diet [30]. Furthermore, our study demonstrates that replacement with GLP-1 receptor agonist (Exendin-4) improves hepatic insulin sensitivity in NSAID-treated mice by having a similar glucose appearance rate to the control group but at a lower insulin level. Interestingly, the GLP-1 receptor knock-out mice were also protected from insulin resistance (specifically hepatic insulin activity) when placed on a high-fat diet [30]. This suggests that the effect of GLP-1 on hepatic glucose trafficking through insulin action might not be through its canonical GLP-1 receptor or through its active metabolite, which is the GLP-1 (7–36) amide. In fact, secondary GLP-1 metabolites, GLP-1 (9–36) amide and GLP-1 (28–36) amide, were also shown to have a positive effect on glucose homeostasis, specifically hepatic glucose production [31,32]. Another study supporting our findings showed that central administration of GLP-1 improves hepatic insulin action and insulin secretion in mice fed with a high-fat diet [33]. Furthermore, a previous study has shown before that administration of indomethacin protects mice from diet-induced obesity but does not improve glucose tolerance, and this process seems to be dependent on GPR40-sigaling [34].

Another recent study by Sugimura et al. showed that the dysbiosis associated with HFD exacerbates the small intestinal injury which is induced by NSAIDs [35].

It is interesting to note that proton pump inhibitors—an increasingly prescribed medication to treat acid reflux as well as gastric and duodenal ulcers—were shown to actually exacerbate NSAID-induced enteropathy [36]. Remarkably, the increase in type 2 diabetes and obesity in the United States paralleled these growth patterns of increase in NSAID use followed by proton pump inhibitors [37]. NSAID-induced enteropathy has been increasingly recognized as a serious health condition with variable clinical presentations affecting a large realm of the population [38]. However, unlike peptic ulcer disease, there is no current medication available to treat it at the moment and no easy endoscopic access to control any bleeding condition [38]. In addition, there seems to be suggestive evidence that NSAID-induced enteropathy might affect glucose metabolism via alteration of GLP-1 secretion. Our study is of course limited by many factors, including testing the actual effect of the GLP-1 effect on hepatic insulin sensitivity by administering central injections in weight-matched animals. In addition, more suggestive data are needed from human subjects to support these findings.

## 5. Conclusions

We postulate that the long-term use of NSAIDs may subsequently contribute to or even lead to further expanding the growing pandemic of type 2 diabetes, nonalcoholic fatty liver disease (NAFLD), and obesity by affecting GLP-1 production as a first step and subsequently hepatic insulin activity and glucose production. Further studies and evaluations in humans are needed to test these hypotheses. Therefore, we conclude that, at least in mice, chronic NSAID administration (in the setting of NSAID-induced enteropathy) causes a state of functional GLP-1 deficiency that is associated with hepatic insulin resistance and is reversible after replacement with a GLP-1 receptor agonist.

## Figures and Tables

**Figure 1 nutrients-14-00120-f001:**
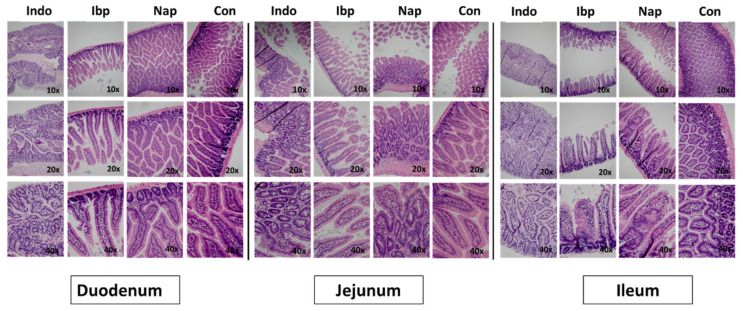
Histological effect of variable NSAIDs on multiple sections of the small intestine. Hematoxylin and Eosin (H&E) staining of duodenum, jejunum and ileum (at variable magnifications 10×, 20×, and 40×) in C57 Blc/6J male mice treated with indomethacin (Indo), ibuprofen (Ibp), naproxen (Nap), or PBS (Con) for 6 weeks. *n* = 5–6.

**Figure 2 nutrients-14-00120-f002:**
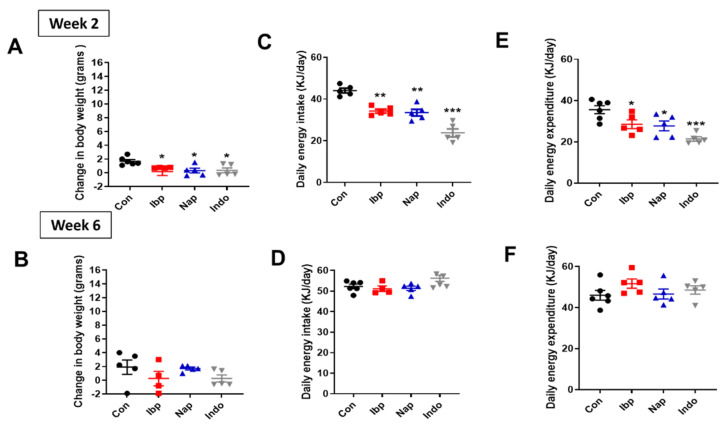
Intragastric administration of variable NSAIDs causes a transient reduction in body weight and alteration in energy balance in normal chow-fed mice. (**A**,**B**) Change in body weight (grams), (**C**,**D**) Average daily energy intake (KJ/day) and (**E**,**F**) daily energy expenditure (KJ/day) during week 2 and week 6 post-intragastric gavage of PBS/control (Con), ibuprofen (Ibp), naproxen, (Nap), or indomethacin (Indo) in C57 Blc/6J male mice that were placed on a normal chow diet (ND). All results are presented as mean ± SEM (error bars). One-way ANOVA was used to compared means of each NSAID group to the control group. ANCOVA correction for body weight was applied for total daily energy expenditure. Statistical significances are denoted with asterisks as follows: *, *p* ≤ 0.05; **, *p* ≤ 0.01; ***, *p* ≤ 0.001.

**Figure 3 nutrients-14-00120-f003:**
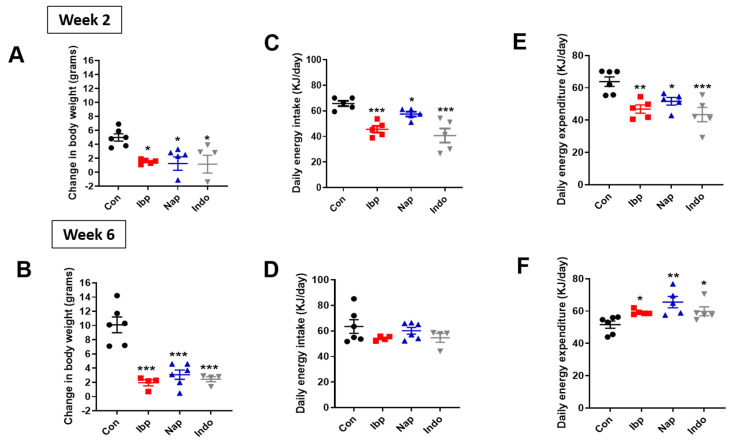
Prolonged intragastric administration of variable NSAIDs protects mice from HFD-induced weight gain. (**A**,**B**) Change in body weight (grams), (**C**,**D**) Average daily energy intake (KJ/day) and (**E**,**F**) total daily energy expenditure (KJ/day) during week 2 and week 6 post-intragastric gavage of PBS/control (Con), ibuprofen (Ibp), naproxen, (Nap), or indomethacin (Indo) in C57 Blc/6J male mice placed on a high-fat diet (HFD). All results are presented as mean ± SEM (error bars). One-way ANOVA was used to compared means of each NSAID group to the control group. *n* = 6 (Con), 4 (Ibp), 5–6 (Nap), and 4–7 (Indo). ANCOVA correction for body weight was applied for total daily energy expenditure. Statistical significances are denoted with asterisks as follows: *, *p* ≤ 0.05; **, *p* ≤ 0.01; ***, *p* ≤ 0.001.

**Figure 4 nutrients-14-00120-f004:**
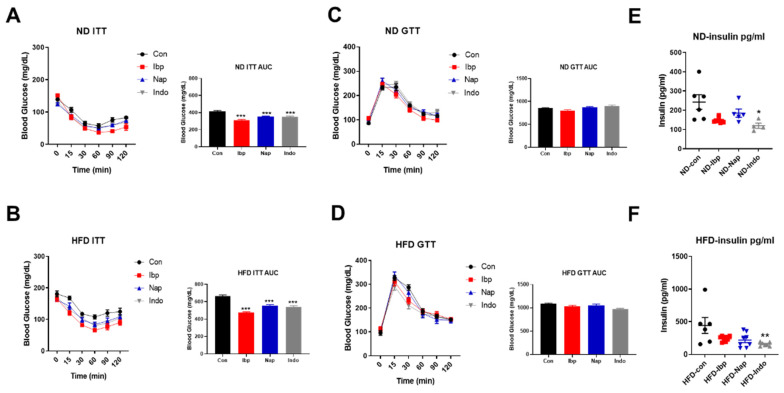
Chronic NSAIDs treatment in mice induces improvement in insulin but not glucose tolerance. Insulin tolerance test (ITT), glucose tolerance test (GTT), and fasting plasma insulin levels in C57 Blc/6J male mice placed on either (**A**,**C**,**E**) normal chow diet (ND) or (**B**,**D**,**F**) high-fat diet (HFD) and treated with PBS (Con), ibuprofen (Ibp), naproxen, (Nap), or indomethacin (Indo) afterr 3–4 weeks. *n* = 6 (Con), 11–12 (Ibp), 6–10 (Nap), and 5–8 (Indo). All results are presented as mean ± SEM (error bars). One WAY-ANOVA was used to compare the differences between control and the three NSAID groups for GTT and ITT. Student’s *t* test was used to compared means between controls and each NSAID group. Statistical significances are denoted with asterisks as follows: *, *p* ≤ 0.05; **, *p* ≤ 0.01; ***, *p* ≤ 0.001.

**Figure 5 nutrients-14-00120-f005:**
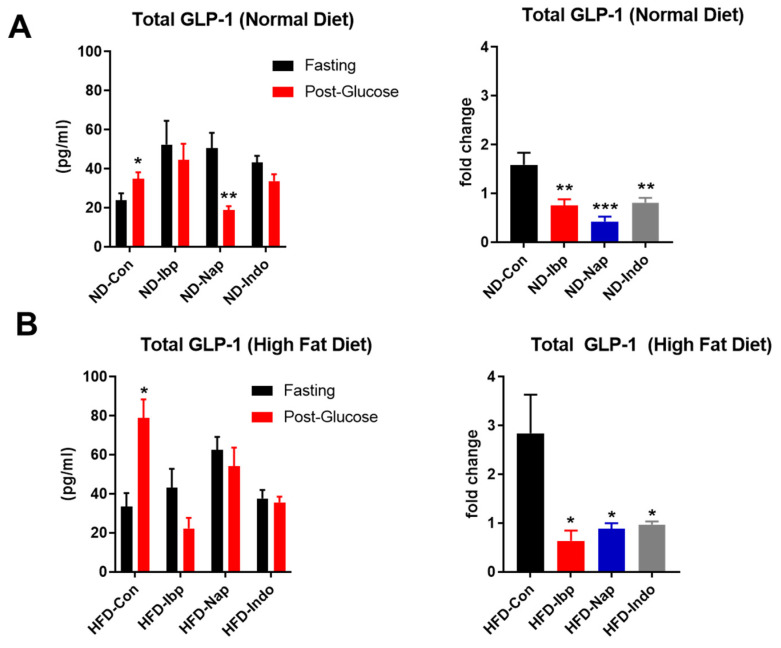
NSAIDs supplementation impairs glucose-stimulated GLP-1 secretion. Serum levels of total GLP-1 expressed in (pg/mL) during fasting (Black bars) and 5 min post-glucose gavage (Red bars) and Fold change in GLP-1 after glucose challenge in C57 Blc/6J male mice placed on either (**A**) a normal chow diet (ND) or (**B**) a high-fat diet (HFD) and treated with PBS (Con), ibuprofen (Ibp), naproxen, (Nap), or indomethacin (Indo) for 4 weeks; *n* = 6 (Con), 4 (Ibp), 6 (Nap), and 5 (Indo). All results are presented as mean ± SEM (error bars) (*n* = 4–8). Student’s *t* test was used to detect differences in GLP_1 levels between fasting and post-glucose gavage states. One-way ANOVA was used to detect differences between control and the three NSAID groups. Statistical significances are denoted with asterisks as follows: *, *p* ≤ 0.05; **, *p* ≤ 0.01; ***, *p* ≤ 0.001.

**Figure 6 nutrients-14-00120-f006:**
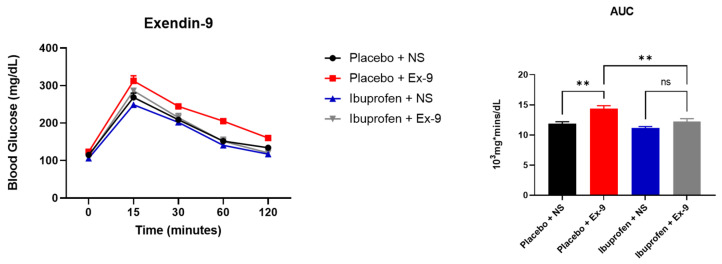
Treatment with GLP-1 receptor antagonist (Exendin-9) worsens glucose tolerance test of placebo but not ibuprofen-treated mice. Glucose tolerance test (GTT) expressed in (mg/dL) in C57Bc/6J mice and treated with placebo vs. ibuprofen for two weeks, followed by the administration of 5 µg of Exendin-9 IP or normal saline 30 min before the experiment: *n* = 5 (Placebo + NS), 5 (Placebo + Ex-9), 6 (ibuprofen + NS), and 6 (ibuprofen + Ex-9). All results are presented as mean ± SEM (error bars) (*n* = 5–6). One-way ANOVA was used to detect differences between control and the three experimental groups. Statistical significances are denoted with asterisks as follows:; **, *p* ≤ 0.01.

**Figure 7 nutrients-14-00120-f007:**
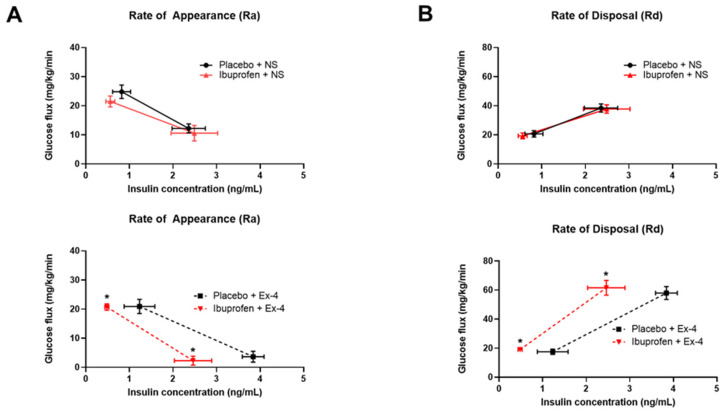
GLP-1 agonist (Exendin-4) improves insulin-mediated glucose appearance and disposal in mice treated with ibuprofen. C57 Blc/6J male mice were placed on a high-fat diet (HFD) and treated with PBS (Placebo) or ibuprofen (Ibp) for 2 weeks, and then a daily injection of (Exendin-4) was administered IP at (24 nmol/Kg) vs. normal saline (NS) for 5 days before undergoing the hyperinsulinemic-euglycemic clamp experiment. The (**A**) glucose appearance rate and (**B**) glucose disposal rate expressed in (mg/kg/min) in relation to the insulin levels expressed in (ng/mL). *n* = 8 (Placebo + NS), 6 (Placebo + Ex-4), 8 (ibuprofen + NS), and 6 (ibuprofen + Ex-4). All results are presented as mean ± SEM (error bars). Student’s *t* test was used to compare means between the (Ex-4) treated groups. Statistical significances are denoted with asterisks as follows: *, *p* ≤ 0.05.

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
