# Peer review of "NSAID-Induced Enteropathy Affects Regulation of Hepatic Glucose Production by Decreasing GLP-1 Secretion"

_nutrients, 2021, doi:10.3390/nu14010120_

Round 1

Reviewer 1 Report

It is a fascinating topic, and I enjoyed the reading. The article is well written, and the study has a good design.
However, there are some spacing, grammatical, and stylistic errors in the article. ex; line 68, 166, 174, 179 so on....
There are many animal research designs, so it makes me confused. It would be great to add a figure for the animal study designs.
It would be clearer to add detailed information on applying NSAID in the method section. How many times per day? one or two?? Put the reference for the NSAID dosage that you used. 
Line 143: what is the post-surgery means? You did not mention any surgery before. Please clarify it. 
Line 151: If you want to explain the process of ELISA in detail, please give more information; otherwise, put references for the protocol. For example, what were the incubation time, the sample volume, how many samples did you run duplicate or more??. .so on
In my opinion, this review paper can be recommended for publication after minor revision.

Author Response

It is a fascinating topic, and I enjoyed the reading. The article is well written, and the study has a good design. However, there are some spacing, grammatical, and stylistic errors in the article. ex; line 68, 166, 174, 179 so on....

We appreciate the reviewer’s comment about our study’s findings. We also appreciate the notification regarding the few unintended errors in typing, spacing and style. We revised the manuscript accordingly to correct all the typing and punctuation mistakes. We also rewrote few of the sentences that we judged as unclear or could be misinterpreted.

There are many animal research designs, so it makes me confused. It would be great to add a figure for the animal study designs.

We apologize for this unintended confusion. First, we clarified the total number of animal in the animal design section in the “Materials and Methods”. In addition, we provided details about the “N” of each experiment group in legend of each figure with statistics involved.  
It would be clearer to add detailed information on applying NSAID in the method section. How many times per day? one or two?? Put the reference for the NSAID dosage that you used. 

We also mentioned that the NSAID administration was every other day in the Method section as well as in the result section. We tested before the daily dose and we found that it was very stressful for the mice and actually caused some fatalities.
Line 143: what is the post-surgery means? You did not mention any surgery before. Please clarify it. 

We do apologize for the mistake in Line 143 that was meant to be post initiation of NSAID treatment. Finally, we provided a visual abstract design that would summarize and simplify our findings as well.

Line 151: If you want to explain the process of ELISA in detail, please give more information; otherwise, put references for the protocol. For example, what were the incubation time, the sample volume, how many samples did you run duplicate or more??. .so on
In my opinion, this review paper can be recommended for publication after minor revision.

We did describe the Insulin ELISA Process in details as performed and as suggested by the producer : https://www.crystalchem.com/ultra-sensitive-mouse-insulin-elisa-kit.html.

Sandwich type immunoassay ELISA is used to measure plasma insulin levels (Crystal Chem, Cat# 90080). The microplate consists of 96-well coated with a monoclonal antibody specific for insulin. In addition to the samples, a detection antibody is added to the microplate wells. The first incubation is completed for 2 hours at 4 °C on a microplate shaker at 700-900 rpm.  Then, the wells were washed using a wash buffer and blotted dry. A conjugation solution of 100 µL were added, followed by a 30 min incubation and another wash. After the addition of 100 µL TMD substrate, the microplate is incubated a second time for 40 mins on a microplate shaker at 700-900 rpm. Finally, 100 µL of stop solution is added followed by measurement of the optical density (OD) by a spectrophotometer at 450 nm. The amount of insulin in each sample is directly proportional to the intensity of the color generated.

Reviewer 2 Report

This paper written by Herz et al. presents a mouse model of NSAID-induced enteropathy and demonstrates that chronic NSAID use causes a functional GLP-1 deficiency associated with hepatic insulin resistance. This state is reversible after replacement with a GLP-1 receptor agonist.

The manuscript is, in general, well written, following the correct steps to present the importance of the problem, the methods used, results and discussions. The results are well presented, including figures for a better understanding of all study steps. The conclusion of the study is supported by the result presented. 

The authors should make some improvements:

  • I recommend not to use abbreviation in the title or key-words
  • the Abstract should be improved, adding the aim of the study, presenting better the methods; abbreviated word GLP-1 in the abstract must be explained
  • all the abbreviated words should be explained at their first use in the paper
  • In the Introduction, at the end of the presentation of the aim of this study must be improved; all the Introduction section is included in only one paragraph; better to separate it 2-3 paragraphs based on ideas presented.
  • In the study design, it would be better to specify the number of mice used for each group; line 87, please explain Jackson laboratory, where? Country? Why brackets?
  • The Discussion section would be better organized in different paragraphs based on the ideas discussed. The authors should emphasize the importance of their results in correlation with the actual clinical situation of NSAID use and obesity and T2DM.
  • The last lines from Discussions could be merged into Conclusions.
  • The authors may add a paragraph regarding the strengths and limitations of their study.
  • Some typos should be corrected (see united states without initial capital letters in line 417)
  • The references include only 6 papers published during the last 5 years; adding more recent studies in discussing this study results would improve the significance and the quality of discussions.

Author Response

This paper written by Herz et al. presents a mouse model of NSAID-induced enteropathy and demonstrates that chronic NSAID use causes a functional GLP-1 deficiency associated with hepatic insulin resistance. This state is reversible after replacement with a GLP-1 receptor agonist.

The manuscript is, in general, well written, following the correct steps to present the importance of the problem, the methods used, results and discussions. The results are well presented, including figures for a better understanding of all study steps. The conclusion of the study is supported by the result presented. 

The authors should make some improvements:

I recommend not to use abbreviation in the title or key-words

We appreciate the reviewer’s advice, but it is honestly difficult in this situation where two important key words NSAID and GLP-1 need to be expressed then the title will be pretty lengthy. We did; however, shorten the title to : “NSAID- induced enteropathy affects hepatic glucose production control by decreasing GLP-1 secretion”.

the Abstract should be improved, adding the aim of the study, presenting better the methods; abbreviated word GLP-1 in the abstract must be explained.

We appreciate the reviewer for the mentioned comment, and we do agree with suggestion. We revised our abstract to the following: Background/Aim: Given their widespread use and their notorious effects on the lining of the gut cells including the enteroendocrine cells, we explored if chronic exposure to Non-steroidal anti- inflammatory Drug (NSAID)s affects metabolic balance in a mouse model of NSAID-induced enteropathy Method: We administered through intragastric gavage variable NSAIDs to C57Blk/6J mice and measured their energy balance, glucose hemostasis, and GLP-1 levels. We treated them with Exendin-9 and Exendin-4 and ran a hyperinsulinemic-euglycemic clamp. Results: Chronic administration of multiple NSAIDs to C57Blk/6J mice induces ileal ulcerations and weight loss in animals consuming high fat diet. Despite losing weight, NSAID-treated mice exhibit no improvement in their glucose tolerance. Furthermore, glucose-stimulated (Glucagon- like peptide -1) GLP-1 is significantly attenuated in the NSAID-treated groups. In addition, Exendin-9 -a GLP-1 receptor antagonist- worsens glucose tolerance test in the control group but not the NSAID-treated ones. Finally, hyper-insulinemic euglycemic clamp study shows that endogenous glucose production, total glucose disposal and their associated insulin levels were similar among an ibuprofen-treated group and its control. Exendin-4, a GLP-1 receptor agonist, reduces insulin levels in the ibuprofen group compared to their controls for the same glucose exchange rates. Conclusion: Chronic NSAID use can induce small intestinal ulcerations which can affect intestinal GLP-1 production, hepatic insulin sensitivity and consequently hepatic glucose production.

all the abbreviated words should be explained at their first use in the paper

Thank you. all abbreviated words were revised in the manuscript and were all explained in their first appearance.

In the Introduction, at the end of the presentation of the aim of this study must be improved; all the Introduction section is included in only one paragraph; better to separate it 2-3 paragraphs based on ideas presented.

We appreciate the suggestion of the reviewer. We, as such divided the introduction into 3 separate paragraphs.

In the study design, it would be better to specify the number of mice used for each group; line 87, please explain Jackson laboratory, where? Country? Why brackets?

We did specify the number of the animals and details about Jackson laboratory in the study design.

Reviewer 3 Report

This paper aims to evaluate the effect and mechanism of chronic NSAIDs treatment on metabolic balance in C57black/6J mice. The manuscript is well written and the information is presented in an orderly manner, and result is interesting. Some minor comments are provided below.

Abstract needs to improve. Method part is more like the purpose of the study and should be in background and a clear method used for this study should be briefly described.

Study Design, Line 90-91 “Each diet group was divided into several NSAID-receiving groups” clarify “several”; how many animals in each group? Line 109 correct “-80o”.

Line 217 Correct “F2.13”

Line 228 it states “for two weeks” but in Figure 1 line 295 “for 6 weeks”.

Author Response

Reviewer 3

This paper aims to evaluate the effect and mechanism of chronic NSAIDs treatment on metabolic balance in C57black/6J mice. The manuscript is well written and the information is presented in an orderly manner, and result is interesting. Some minor comments are provided below.

Abstract needs to improve. Method part is more like the purpose of the study and should be in background and a clear method used for this study should be briefly described.

We thank the reviewer for the comment. We did indeed revise our abstract describing our study design briefly in the new version.

Study Design, Line 90-91 “Each diet group was divided into several NSAID-receiving groups” clarify “several”; how many animals in each group? Line 109 correct “-80o”.

We thank  the reviewer for the notification. We corrected the typos and we also clarified in the revised manuscript the number of mice used in each experiment.

Line 217 Correct “F2.13”

We did correct the punctuation mistake. Thank you.

Line 228 it states “for two weeks” but in Figure 1 line 295 “for 6 weeks”.

We apologize for the confusion. However, these are two separate experiments. The GTT was a one-time study done with Exendin-9 after 2 weeks  vs controls to look at the effect of GLP-1 receptor on GTT. The other GTTs are all done between week 3-4 during a 6-weeks gavage experiment. We revised our Methods to clarify each experiment accordingly.